# Cell-Free Nuclear and Mitochondrial DNA as Potential Biomarkers for Assessing Sepsis Severity

**DOI:** 10.3390/biomedicines12050933

**Published:** 2024-04-23

**Authors:** Felipe Silva de Miranda, Livia Maria A. M. Claudio, Dayanne Silva M. de Almeida, Juliana Braga Nunes, Valério Garrone Barauna, Wilson Barros Luiz, Paula Frizzera Vassallo, Luciene Cristina Gastalho Campos

**Affiliations:** 1Department of Biological Science State, University of Santa Cruz, Ilhéus 45662-900, Bahia, Brazil; felipemiranda2004@hotmail.com (F.S.d.M.); almeida.dsm@gmail.com (D.S.M.d.A.); ju_braga@hotmail.com (J.B.N.); wbluiz@uesc.br (W.B.L.); 2Postgraduate Program in Biology and Biotechnology of Microorganisms State, University of Santa Cruz, Ilhéus 45662-900, Bahia, Brazil; 3Laboratory of Applied Pathology and Genetics State, University of Santa Cruz, Ilhéus 45662-900, Bahia, Brazil; 4Post Graduation Program in Physiological Sciences, Federal University of Espírito Santo, Vitória 29075-910, Espírito Santo, Brazil; livia.ama@hotmail.com; 5Molecular Physiology Laboratory of Exercise Science, Federal University of Espírito Santo, Vitória 29075-910, Espírito Santo, Brazil; barauna2@gmail.com; 6Clinical Hospital, Federal University of Minas Gerais, Belo Horizonte 30130-100, Minas Gerais, Brazil; pfvassallo1@gmail.com

**Keywords:** cfDNA, circulating nucleic acids, intensive care unit (ICU), biomarkers, infectious diseases

## Abstract

Sepsis continues to be a significant public health challenge despite advances in understanding its pathophysiology and management strategies. Therefore, this study evaluated the value of cell-free nuclear DNA (cf-nDNA) and cell-free mitochondrial DNA (cf-mtDNA) for assessing the severity and prognosis of sepsis. Ninety-four patients were divided into three groups: infection (n = 32), sepsis (n = 30), and septic shock (n = 32). Plasma samples were collected at the time of diagnosis, and cfDNA concentrations were determined by qPCR assay. The results showed that plasma cfDNA levels increased with the severity of the disease. To distinguish between patients with infection and those with sepsis, the biomarker L1PA2_90_ achieved the highest AUC of 0.817 (95% CI: 0.725–0.909), demonstrating a sensitivity of 77.0% and a specificity of 79.3%. When cf-nDNA was combined with the SOFA score, there was a significant improvement in the AUC (0.916 (0.853–0.979)), sensitivity (88.1%), and specificity (80.0%). Moreover, patients admitted to the ICU after being diagnosed with sepsis had significantly higher cf-nDNA concentrations. In patients admitted to the ICU, combining cf-nDNA with the SOFA score yielded an AUC of 0.753 (0.622–0.857), with a sensitivity of 95.2% and a specificity of 50.0%. cfDNA can differentiate between patients with infection and those with sepsis. It can also identify patients who are likely to be admitted to the ICU by predicting those with indications for intensive care, suggesting its potential as a biomarker for sepsis.

## 1. Introduction

Sepsis is a potentially life-threatening condition characterized by organ dysfunction resulting from a dysregulated immune response to infection [1]. It is estimated that 49 million cases of sepsis occur worldwide each year, resulting in 11 million sepsis-related deaths. This condition is considered a severe public health problem [2]. Additionally, it is the primary cause of death in non-cardiac intensive care units (ICUs), with mortality rates that differ based on the socioeconomic characteristics of the country [3].

Early identification and diagnosis of organ dysfunction in patients with infection are directly related to their prognosis [4]. Therefore, treatment should ideally be initiated within the first hour following the diagnosis of sepsis or septic shock. Each hour of delay in treatment could increase the risk of dying from sepsis by up to 8% [5]. Actually, an increase of 2 points in the Sequential Organ Failure Assessment (SOFA) score indicates possible organ dysfunction and is used for sepsis diagnosis. Septic shock is considered if a patient has persisting hypotension requiring vasopressors to maintain a mean arterial pressure (MAP) ≥ 65 mm Hg and a serum lactate level > 2 mmol/L, despite adequate volume resuscitation [1,6,7].

Additionally, the most commonly used biomarkers in clinical practice—C-reactive protein (CRP), procalcitonin (PCT), and lactate—have limited sensitivity and specificity for diagnosis and determining severity. The SOFA score has higher sensitivity compared to these biomarkers, but its specificity is limited because various non-infectious conditions can also lead to organ dysfunction [6,8,9]. Therefore, in the absence of a gold standard method [1], it is crucial to assess potential biomarkers that can diagnose sepsis and identify patients with more severe conditions [10]. In this context, molecular biomarkers such as cell-free DNA (cfDNA) are being investigated in sepsis and various other pathologies as potential biomarkers and noninvasive screening tools. These include cancer, trauma, myocardial infarction, stroke, transplant patient monitoring, and prenatal screening for genetic abnormalities [11].

The relationship between cfDNA and various pathologies can be evaluated by examining the molecular characteristics of cfDNA, including plasma concentration, integrity, and cellular origin, among other factors, to determine the severity of these diseases. The significant increase in plasma cfDNA levels may be associated with pathological processes, such as immune and inflammatory responses, as well as organ dysfunction. The integrity of cfDNA is assessed by analyzing fragment sizes, which reflect the underlying cellular pathophysiological conditions. Necrosis is associated with larger fragment sizes, whereas apoptotic release results in shorter fragments. Additionally, the assessment of cell-free mitochondrial DNA (cf-mtDNA) may serve as a more sensitive diagnostic tool compared to cell-free nuclear DNA (cf-nDNA), which is attributed to the lack of nucleosome-associated histone proteins in mitochondria, which leads to the presence of shorter and more abundant fragments in the plasma [11]. All these molecular features, when combined, can serve as assessment tools for septic patients, just as they are currently used in clinical practice in obstetrics and oncology. Early diagnosis and stratification of the severity of this syndrome increase the possibilities for interventions that could have an impact on mortality.

Therefore, this study aimed to assess cell-free nuclear DNA (cf-nDNA) and cell-free mitochondrial DNA (cf-mtDNA) as potential tools for predicting the severity of sepsis. Furthermore, we investigated whether the increased copy numbers of cf-nDNA and cf-mtDNA are associated with ICU admissions in patients with sepsis.

## 2. Materials and Methods

### 2.1. Overview

This is an observational, prospective, cross-sectional study conducted at a public university hospital in the capital of Espírito Santo state, Brazil, from March 2017 to December 2018. The study was submitted and approved by the ethical committee of the Federal University of Espírito Santo under the number 2889-718. All research was performed in accordance with the relevant guidelines and regulations of the Brazilian National Health Council (CNS) Resolution 466/2012 [12] and the Declaration of Helsinki [13]. Patients admitted to the ICU, emergency department, and internal medicine were included. Diagnoses of infection, sepsis, and septic shock were made according to Sepsis-3 consensus definitions. The qSOFA and SOFA scores were calculated according to the criteria described by Sepsis-3 [1,6]. Out of a total of one hundred and thirty-one (131) patients enrolled consecutively in the study through convenient nonprobability sampling, ninety-four (94) adult patients diagnosed with infection, sepsis, or septic shock were included, while thirty-seven (37) patients were excluded due to missing data in medical records, such as the absence of laboratory results, SOFA scores, and outcome information (Figure 1). The control group consisted of patients diagnosed with infection, while the case group comprised those diagnosed with sepsis or septic shock. Written informed consent was provided to all patients. Pediatric patients (under 18 years of age), individuals in a palliative care state, immunocompromised individuals, pregnant women, those who declined to provide consent for study participation, and subjects for whom a sepsis diagnosis could not be confirmed were excluded from this study. Blood samples were collected within 30 min after diagnosis and before the initiation of antibiotic treatment. Four milliliters of peripheral blood was collected in BD Vacutainer tubes containing EDTA (BD, Franklin Lakes, NJ, USA) and centrifuged at 2000× *g* for 10 min at room temperature. The plasma was transferred to a microcentrifuge tube and stored at −80 °C for subsequent analysis. The study was reviewed and approved by the Ethics Committee of the Federal University of Espírito Santo under approval number 2889-718. Written informed consent was provided to patients. All research was conducted in compliance with Brazilian human research legislation, specifically the National Health Council (CNS) Resolution 466/2012 [12], and the Declaration of Helsinki [13].

### 2.2. Clinical Data

Data were collected from the medical records of all patients, including demographic and laboratory data, diagnoses, comorbidities, in-hospital mortality rates, and SOFA and quick SOFA (qSOFA) scores.

### 2.3. Analytical Assays

cfDNA was extracted from plasma by the phenol-chloroform method [14]. Briefly, 0.25 mL of plasma in each sample was extracted. The final elution volume was 50 µL. qPCR was performed on QuantStudio™ 3 Real-Time PCR System (Thermo Fisher Scientific, Waltham, MA, USA). For the quantification of cell-free nuclear DNA targeting L1PA2_90_, sequences of the following primer pair were used: Forward (5′-TGCCGCAATAAACATACGTG-3′) and reverse (5′-GACCCAGCCATCCCATTAC-3′); and for targeting L1PA2_222_, sequences of the following primer pair were used: Forward (5′-TGCCGCAATAAACATACGTG-3′) and reverse (5′-AACAACAGGTGCTGGAGAGG-3′) [15]. For the quantification of cell-free mitochondrial DNA, we used a set of primers: Forward (5′-CTATCCGCCATCCCATACATTG-3′) and reverse (5′-ATCGTGTGAGGGTGGGACTG-3′) targeting mitochondrial genome (Mt:15195-15279) with an amplicon of 85 bp [16]. The qPCR conditions were as follows: an initial incubation for 5 min at 95 °C, followed by 40 cycles of 15 s at 95 °C and 1 min at 60 °C. PCR reactions were performed in triplicate using 2 µL of isolated DNA, 5 µL of SYBR Green (Thermo Fisher Scientific, Waltham, MA, USA), 0.1 µL each of forward and reverse primers, and 2.9 µL of DEPC-treated water to reach a final volume of 10 µL. Negative controls (NTCs) were used, each containing 2 µL of DEPC-treated water.

The concentration of cf-nDNA was calculated using the following equation:$$nDNA=\left(\frac{c}{3.3}\right)*\left(\frac{Velution}{Vplasma}\right)$$

In the equation above, nDNA is the cf-nDNA copy number per milliliter, “c” nDNA concentration (pg/µL) determined by qPCR targeting the nuclear gene L1PA290 or L1PA2222 sequence, and 3.3 pg is the human haploid genome mass. Velution is the volume of cirDNA extract (µL) and Vplasma is the volume of plasma used for the extraction (ml) [17].

Abbreviations: c: cf-nDNA concentration in pg/μL; Velution: elution volume of extracted DNA in μL; Vplasma: volume of plasma used for extraction in mL [17].

cf-mtDNA was calculated based on the following equation:$$mtDNA=\left(\frac{c*NA\;(6.02\;*\;10^{23})}{2*L*MW}\right)*\left(\frac{Velution}{Vplasma}\right)$$

In the equation above, mtDNA is the cf-mtDNA copy number per milliliter, “c” is the mtDNA mass concentration (g/µL) determined by a qPCR targeting the mitochondrial gene. NA is Avogadro’s number (6.02 × 10^23^ molecules per mole), “L” is the mitochondrial amplicon length (85 pb), and MW is the molecular weight of one nucleotide (g/mol). V_elution_ is the elution volume of cirDNA extract (µL) and V_plasma_ is the volume of plasma used for the extraction (mL) [17].

Abbreviations: c: cf-nDNA concentration in g/µL; Velution: elution volume of extracted DNA in μL; Vplasma: volume of plasma used for extraction in mL [17]. After the equation, the values are converted to a base-2 logarithm. cfDNA quantities are expressed as log2 copy numbers/mL.

### 2.4. Statistical Analysis

SPSS Statistics version 23 (International Business Machines, New York, NY, USA) and MedCalc version 20.214 (MedCalc Software Ltd., Ostend, Belgium) were used to perform the statistical analyses. A normality test, the Kolmogorov–Smirnov test, was carried out on all quantitative variables. For demographic, clinical, and laboratory characteristics, categorical data were presented as relative frequencies, and quantitative data were presented as medians with interquartile ranges. The data were stratified by diagnosis and compared using Fisher’s Exact Test or the Kruskal–Wallis H Test. The results of cfDNA quantification were expressed as log2 copy numbers/mL. The Mann–Whitney U test was used to compare cfDNA levels between the different groups. The Spearman rank correlation coefficient was used to assess the relationship between cfDNA levels and SOFA scores. The predictive performance of cfDNA levels was assessed using ROC analysis. The cut-off point was calculated from the Youden index. The area under the curve (AUC) was compared by the DeLong test. Binary logistic regression was adjusted for comorbidities, age, and sex, showing adjusted odds ratios. A *p*-value < 0.05 was considered statistically significant.

## 3. Results

### 3.1. Patients’ Characteristics

A total of 94 patients were included in the study. Of those, 32 patients were diagnosed with infection (34.0%), 30 with sepsis (31.9%), and 32 with septic shock (34.0%). Patients’ demographic (Table 1), clinical (Table 2), and laboratory (Table 3) characteristics were evaluated according to their diagnosis.

The median age was 65 (48.7–76) years. Most patients were male (53.8%) (Table 1). Most diagnoses were from the internal medicine department (46.5%), followed by the emergency department (33.7%) and the ICU (19.8%). A total of 46.9% of the patients were admitted to the ICU, with the majority being diagnosed with septic shock (70.0%) and sepsis (54.2%) (*p* < 0.0001). The median length of hospital stay was 22 (10–35.5) days. The fatality rate was 28%. As expected, patients with prolonged hospital stays have more severe clinical conditions (*p* = 0.003). Patients with sepsis/septic shock also had higher mortality rates, higher SOFA scores, and higher qSOFA positivity compared to the infection group (*p* < 0.0001). The most common comorbidities identified were hypertension (45.7%), heart disease (28.7%), and diabetes (24.5%). The patients in the septic shock group had a higher incidence of hypertension (71.9%; *p* = 0.001) and cardiopathy (50.0%). In contrast, the group with infections had the highest proportion of patients without comorbidities (37.5%; *p* = 0.006). Healthcare-associated infections accounted for 60.5% of the cases. The most common source of infection was unidentified (31.9%), followed by respiratory (26.6%), genitourinary tract (17.0%), and abdominal infections (14.9%) (Table 2).

Regarding laboratory tests, they were collected within 30 min after the diagnosis was made. There were significant differences in the measurements of INR (*p* = 0.014), total bilirubin (*p* = 0.023), creatinine (*p* = 0.012), and lactate (*p* = 0.030), with levels increasing progressively with the severity of the patient’s condition (Table 3).

### 3.2. cfDNA Concentration according to Sepsis Severity

Plasma levels of cf-nDNA (L1PA2_90_ and L1PA2_222_) and cf-mtDNA were measured and compared among patients with infection, sepsis, and septic shock. The measurements are expressed as log2 values of the number of copies/mL. The results show higher plasma cfDNA levels correlating with disease severity (Figure 2).

The median L1PA2_90_ concentration level was 17.28 (15.78–18.05) in patients with infection, 18.75 (17.63–19.96) in patients with sepsis, and 19.77 (18.48–22.33) in patients with septic shock. There are significant differences between patients with infection and those with sepsis (*p* < 0.0001), as well as between patients with infection and those with septic shock (*p* < 0.0001). Median levels of L1PA2_222_ were 17.60 (15.68–18.11) in the infection group, 17.93 (16.83–19.66) in the sepsis group, and 19.69 (18.52–21.64) in the septic shock group. Significant differences were observed between the infection and shock groups (*p* < 0.0001), as well as between the sepsis and septic shock groups (*p* = 0.001). Lastly, cf-mtDNA levels were measured at 37.08 (35.94–38.18) in patients with infection, 38.27 (36.00–40.01) in patients with sepsis, and 38.62 (37.39–39.85) in patients with septic shock. Significant differences were observed only between patients with infection and those with septic shock (*p* < 0.0001).

### 3.3. Correlation between cfDNA Levels and SOFA Scores

A moderate, positive, and significant correlation was observed between L1PA2_90_ and the SOFA score (rho = 0.577, 95% CI 0.418–0.702, *p* < 0.0001). In contrast, a weak, positive, and significant correlation was found between L1PA2_222_ and cf-mtDNA (rho = 0.454, 95% CI 0.268–0.608, *p* < 0.0001), and between cf-mtDNA and the SOFA score (rho = 0.267, 95% CI 0.0605–0.451, *p* = 0.0121) (Figure 3).

### 3.4. Potential Diagnostic Value of cfDNA

ROC curves were generated to assess the potential of each cfDNA biomarker to distinguish between patients with infection and those with sepsis or septic shock. The results are shown in Figure 4. The ROC curve analysis indicated that L1PA2_90_ achieved the highest AUC, which was 0.817 (95% CI 0.725–0.909), with a sensitivity of 77.0% and a specificity of 79.3% at a cut-off point of 18.07. Followed by L1PA2_222_, which had an AUC of 0.741 (95% CI: 0.634–0.849), with a sensitivity of 66.6% and a specificity of 81.4% at a cut-off point of 18.27 (Table 4). Additionally, binary logistic regression analysis indicates that the concentration of L1PA2_90_ (OR: 2.067 (95% CI 1.449–2.946); *p* < 0.0001), L1PA2_222_ (OR 1.655 (95% CI 1.240–2.208) *p* = 0.001), and cf-mtDNA (OR 1.415 (95% CI 1.122–1.784); *p* = 0.003) were significantly associated with disease severity. When cf-nDNA was combined with the SOFA score, there was a significant improvement in AUC (0.916 [95% CI 0.853–0.979]), sensitivity (88.1%), and specificity (80.0%) compared to previously described results (*p* < 0.05).

### 3.5. cfDNA and ICU Hospitalization

An exploratory analysis was conducted on patients diagnosed in the emergency and internal medicine departments to assess the association between cfDNA levels and ICU admissions. Therefore, 17 patients who were diagnosed directly in the ICU were excluded from this analysis. Patients admitted to the ICU after diagnosis had significantly higher cf-nDNA concentrations than those who remained hospitalized in internal medicine or were discharged from the hospital. This result is not observed with cf-mtDNA, as there are no significant differences in distribution between the groups. The L1PA2_90_ concentrations are significantly higher in the group of patients admitted to the ICU, with levels at 19.12 (17.94–21.53), compared to those not admitted, whose levels were 18.02 (16.42–19.44) (*p* = 0.013). The same pattern is observed for L1PA2_222_, with concentrations of 18.47 (16.91–21.48) in patients admitted to the ICU and 17.52 (15.98–19.07) in those not admitted (*p* = 0.049). However, cf-mtDNA levels showed no significant difference between patients admitted to the ICU, with levels at 38.02 (36.29–40.19), and those who were not hospitalized, with levels at 37.54 (35.80–38.82) (*p* = 0.190) (Figure 5).

ROC curve analysis of L1PA2_90_ (AUC = 0.692, 95% CI: 0.559–0.806) and L1PA2_222_ (AUC = 0.653, 95% CI: 0.520–0.770) obtained the best results in differentiating patients who were admitted to ICU, with a sensitivity of 100.0% and 66.6% and specificity of 34.2% and 62.5%, respectively. The optimal cut-off point used was 17.17 for L1PA2_90_ and 18.11 for L1PA2_222_. cf-nDNA with SOFA score combined obtained an AUC of 0.753 (95% CI 0.622–0.857), with a sensitivity of 95.2% and a specificity of 50.0% (Table 5). Once again, the combination of SOFA score with cf-nDNA improved accuracy (Figure 6).

## 4. Discussion

Our data indicated that cfDNA concentrations were higher by disease severity, and L1PA2_90_ was able to significantly differentiate between patients with infection, sepsis, and septic shock. ROC curve analysis of cf-nDNA demonstrated improved diagnostic accuracy, and combining cf-nDNA with the SOFA score further increased this accuracy. Similarly, in a subgroup analysis, patients who were hospitalized in the ICU had higher cf-nDNA concentrations.

Sepsis biomarkers have been the subject of intense research [18]. Pierrakos and Vincent estimated that at least 178 biomarkers of sepsis had been reported in the literature [19]. These include acute phase proteins such as C-reactive protein, procalcitonin, pro- and anti-inflammatory cytokines; damage-associated molecular pattern (DAMPs) (e.g., calprotectin and NGAL); endothelial cells and blood–brain barrier markers (e.g., occluding and claudin-5); miRNA (e.g., miR-125a, miR-21); hormones and peptide precursors (e.g., Adrenomedullin, NT-proBNP); and many others [18,20]. Identifying clinically relevant biomarkers would facilitate a more accurate determination of the immune and inflammatory status of a septic patient and could significantly contribute to the stratification of patients who may benefit from a certain therapeutic strategy.

There are several studies evaluating the role of cfDNA in critically ill patients, including those with sepsis. Some of these studies have focused on quantifying cfDNA levels to distinguish septic patients compared to healthy controls or to assess whether the marker is associated with mortality. There are many origins for plasma cfDNA, which can be derived from physiological and pathological processes such as necrosis, apoptosis, exogenous sources, and active cell release, among others. Its origin is heterogeneous, stemming from various organs and cell types. Thus, changes in plasma cfDNA concentrations may reflect the presence of a pathological disorder and its characteristics [11]. Cell-free nuclear DNA and cell-free mitochondrial DNA constitute the plasma cfDNA and have distinct characteristics that need to be differentiated between analyses [21]. The average mean size of cf-nDNA is approximately 166 bp, and larger fragments are associated with necrosis [11]. The analyzed regions are derived from LINE-1 sequences that are distributed across all chromosomes and represent approximately 17% of the entire human genome. Its amplification increases the sensitivity of cfDNA measurement. In turn, cf-mtDNA is more fragmented than cf-nDNA, its mean plasma concentrations are higher, and its increase is associated with pro-inflammatory cytokines such as TNF-α and IL-6 [22,23]. Due to its characteristics and origins, cfDNA can indirectly identify the presence of organ dysfunction in patients with infections. Moreover, using more than one cfDNA target can increase biomarker sensitivity and accuracy.

Our results indicated that patients with sepsis and septic shock have higher levels of circulating cfDNA than patients with infections likely due to organ dysfunction. This was particularly evident with L1PA2_90_, which showed significant differences between the infection group and the group of patients with sepsis and septic shock (*p* < 0.0001). Furthermore, it had the highest AUC of 0.817 (95% CI 0.725–0.909).

Clementi et al. evaluated 27 patients with sepsis and septic shock admitted to the ICU and demonstrated that cf-nDNA can serve as a prognostic marker of severity and has the potential to discriminate between septic and non-septic patients. Additionally, higher levels of cfDNA were associated with acute renal failure requiring renal replacement therapy and an increased duration of mechanical ventilation [24]. Duplessis et al. identified a trend in cf-nDNA concentrations, with higher cfDNA concentrations correlating with increased sepsis severity, and an AUC of 0.650 (95% CI 0.440–0.850). However, they found no significant difference between survivors and non-survivors [25]. Rannikko et al., when evaluating 481 patients with infection, found that the combination of qSOFA score with cf-nDNA predicted 7-day mortality, identifying which patients were at high risk of death. The combination of markers enabled the improvement of AUC from 0.730 (95% CI 0.65–0.82) to 0.770 (95% CI 0.68–0.86) [26].

Regarding cf-mtDNA, our results demonstrate smaller differences between the comparison groups and a lower AUC compared to cf-nDNA, despite having higher specificity when compared to other markers, including cf-nDNA combined with SOFA. Timmermans et al. conducted a prospective observational study in which they collected serial blood samples from 121 patients diagnosed with septic shock, up to the 28th day of their hospitalization. They found that cf-nDNA, cf-mtDNA, and cytokine levels were significantly elevated on the day of diagnosis compared to healthy controls and that these levels were maintained throughout the analyzed period. cf-mtDNA concentrations correlated with IL-1RA, noradrenaline dose, mean arterial pressure, and white blood cell count [27]. Schäfer et al. also found significant differences in cf-mtDNA concentrations between patients diagnosed with sepsis and healthy individuals, with a 123-fold increase [28]. Bhagirath et al., in a translational study evaluating the plasma of 12 septic patients admitted to the ICU, identified concentrations of cf-mtDNA that were 50-fold greater than those in healthy controls and 200-fold greater for cf-nDNA. Additionally, it has been found that a high concentration of cf-mtDNA increases neutrophil viability and activates coagulation and platelets, thereby contributing to the pathophysiology of sepsis [29].

Even with many published studies, there is still a knowledge gap in the link between cfDNA and sepsis. Our study was unprecedented in that it was conducted across the department of internal medicine, emergency, and ICU, not just in intensive care, thereby encompassing a more complete patient profile. Moreover, this study not only encompasses a comprehensive range of diagnoses, including the distinction between infection, sepsis, and septic shock, but also sets itself apart from other studies that typically compare septic patients with healthy controls or non-septic cases exhibiting systemic inflammatory response syndrome (SIRS) in the ICU [10]. cfDNA has the potential to be an efficient biomarker for sepsis and may also assist in differentiating between the stages of the disease. Early identification of organ dysfunction in patients with infection is one of the most important steps in sepsis diagnosis. Thus, one important aspect of this research lies in distinguishing between patients with infection and those with sepsis, rather than employing a control group comprising healthy individuals. This differentiation is crucial and holds relevance in clinical practice, offering valuable insights that can inform targeted interventions and enhance the precision of patient care.

We found that cf-nDNA serves as an independent predictor of severity in patients with infections. Consequently, we conducted an exploratory subanalysis to evaluate the association with ICU admission. Although the analysis is based on a small sample, the analysis seems to confirm a relationship between higher cfDNA levels and increased disease severity. Changes in these biomarkers may serve as valuable indicators, facilitating the identification of hospitalized patients who are at a heightened risk and exhibit a more pronounced need for intensive care. Further research in this context is needed. Despite advances in the study of cfDNA as a biomarker, difficulties in standardizing their quantification remain an obstacle to comparing findings between studies and implementing them in clinical practice [30].

One of the limitations present in total cfDNA quantification analysis is that it does not reveal which tissue these DNA fragments come from. Assessments of cfDNA methylation patterns can be used to identify the tissue origin of cfDNA. This restriction is circumvented by choosing appropriate target genes and their epigenetic signature. This target selection is one of the main factors in achieving relevant and accurate clinical significance [31]. This assessment can be extended to assess specific organ dysfunction in sepsis, such as kidney, liver, or heart damage. The greatest challenge in managing sepsis lies in its varied clinical presentations, which complicate the clinical evaluation and treatment of affected patients. Nevertheless, these challenges can be overcome through the utilization of artificial intelligence tools, specifically with the integration of machine learning with clinical data and molecular markers, such as cfDNA. This approach enables the development of clinical decision support tools pertaining to sepsis and septic shock, ultimately enhancing outcomes for clinicians and facilitating real-time optimization of medical resources [32,33].

## 5. Conclusions

In conclusion, our data showed that cfDNA concentrations increase with the severity of the clinical state in sepsis. These trends are more pronounced when comparing cf-nDNA with cf-mtDNA. cf-nDNA can accurately differentiate between patients with infection and those with sepsis or septic shock. There is an increase in accuracy, sensitivity, and specificity with the combination of cf-nDNA and organ dysfunction score, which can assist in sepsis diagnosis. Furthermore, the same occurs when assessing ICU hospitalization. The elevated cf-nDNA levels observed in patients admitted to the ICU following a sepsis diagnosis not only corroborate the severity of their clinical condition but also indicate the potential for cfDNA to serve as an early indicator of the need for escalated care. Overall, cfDNA could assist healthcare professionals in assessing the severity of illness, and prognosis, and in making decisions regarding patients’ admission to the intensive care unit, thereby showing potential clinical applicability.

## Figures and Tables

**Figure 1 biomedicines-12-00933-f001:**
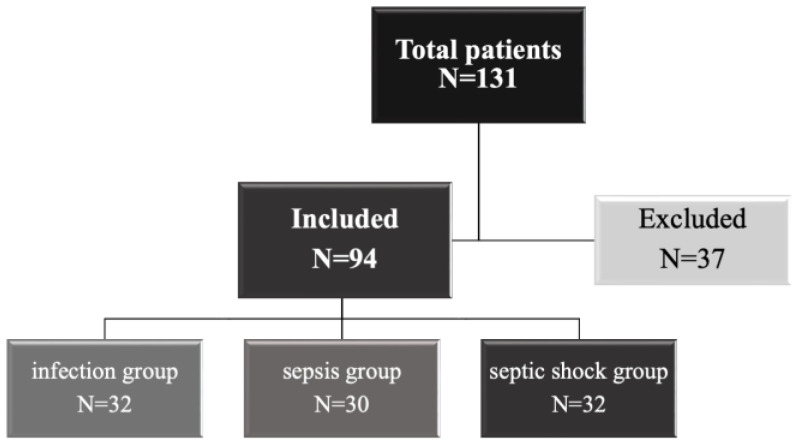
Flow diagram of the patients included in the study.

**Figure 2 biomedicines-12-00933-f002:**
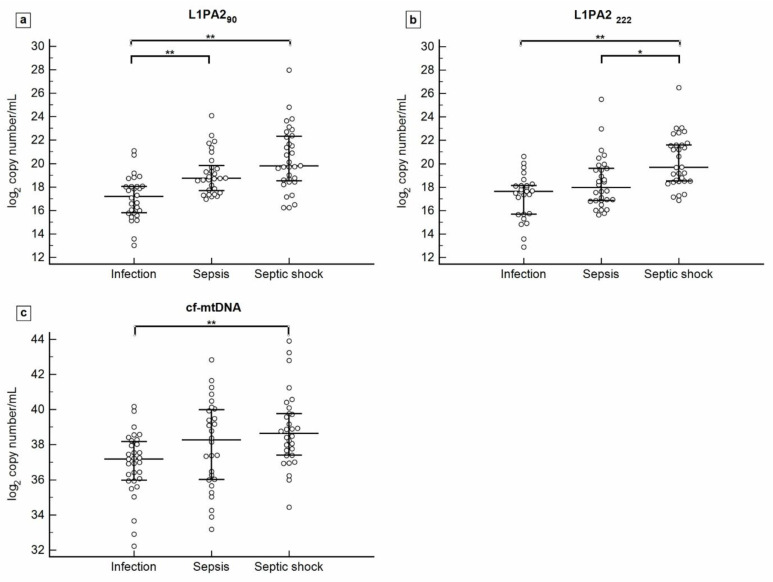
Plasma levels of L1PA2_90_ (**a**), L1PA2_222_ (**b**), and cf-mtDNA (**c**) in patients with infection, sepsis, and septic shock. Data are graphically represented as median and interquartile range (IQR). Abbreviation: * *p* < 0.05, ** *p* < 0.0001.

**Figure 3 biomedicines-12-00933-f003:**
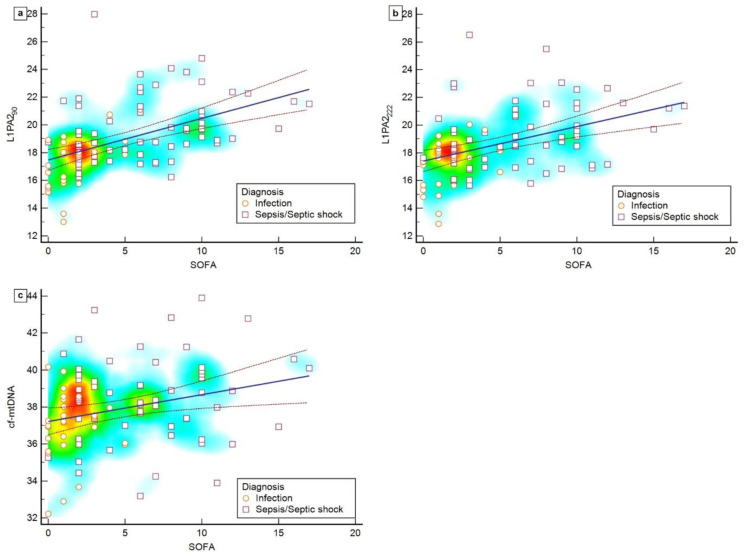
Scatter plot showing the correlation between SOFA score and L1PA2_90_ (**a**), L1PA2_222_ (**b**), and cf-mtDNA (**c**). The orange circle represents patients with infections, while the purple square represents patients with sepsis or septic shock. The colors on the heat map range from blue to red. The higher the density of points, the closer the color approaches red, suggesting cluster formation.

**Figure 4 biomedicines-12-00933-f004:**
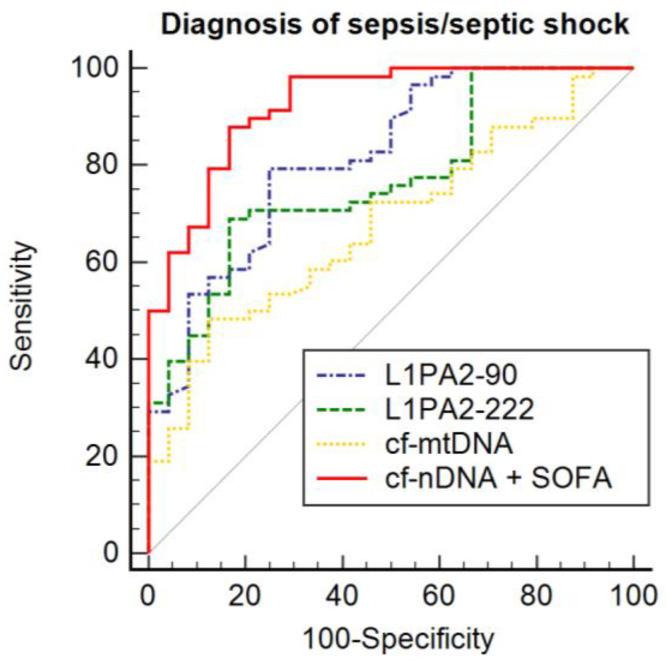
Comparison of ROC curves according to diagnosis.

**Figure 5 biomedicines-12-00933-f005:**
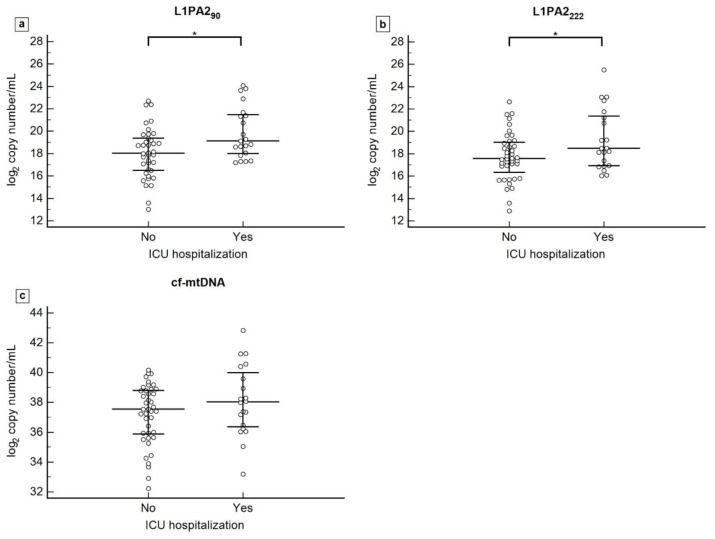
Plasma levels of L1PA2_90_ (**a**), L1PA2_222_ (**b**), and cf-mtDNA (**c**) according to ICU admission. Data are graphically represented as median and interquartile range (IQR). Abbreviation: * *p* < 0.05.

**Figure 6 biomedicines-12-00933-f006:**
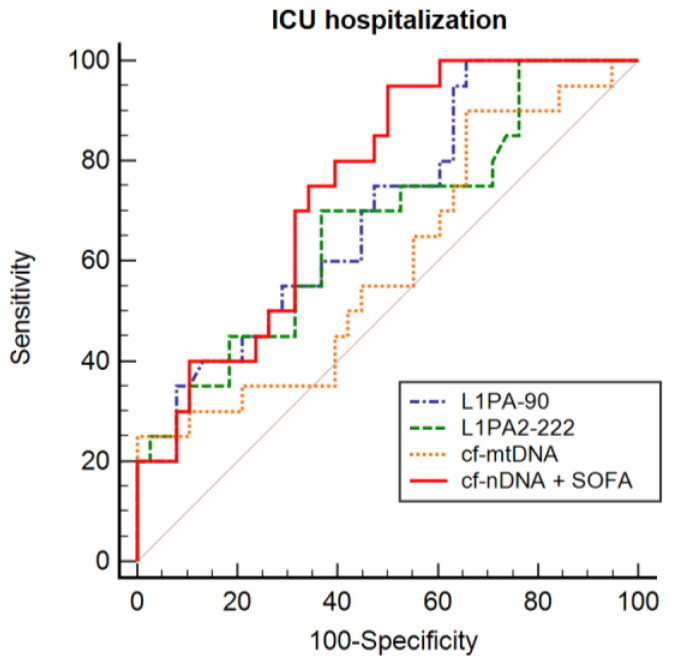
Comparison of ROC curves according to ICU admission.

**Table 1 biomedicines-12-00933-t001:** Demographic characteristics according to diagnosis.

	All Cases (N = 94)	Infection (n = 32)	Sepsis (n = 30)	Septic Shock (n = 32)	*p*-Value †
Age at diagnosis, median (IQR)	**65 (48.7–76)**	54.5 (35–68)	61.5 (49.5–75.2)	75 (68.2–79)	**<0.0001 ***
Male gender (%)	**53.8**	35.5	53.3	71.9	**0.015 ***

Abbreviations: IQR: interquartile range; †: Kruskal–Wallis H test or Fisher’s exact test; *: *p* < 0.05.

**Table 2 biomedicines-12-00933-t002:** Clinical characteristics according to diagnosis.

	All Cases (N = 94)	Infection (n = 32)	Sepsis (n = 30)	Septic Shock (n = 32)	*p*-Value †
Source of diagnosis, (%)					
Emergency	**33.7**	24.0	36.7	38.7	
Internal medicine	**46.5**	72.0	50.0	22.6	0.397
ICU	**19.8**	4.0	13.3	38.7	
ICU hospitalization (%)	**46.9**	14.8	**54.2**	**70.0**	**<0.0001 ***
Length of stay, median (IQR)	**22 (10–35.5)**	11 (7–23.5)	23.5 (10.2–36.0)	26 (13.5–41)	**0.003 ***
SOFA, median (IQR)	3 (2–8)	1 (0–2.2)	**4 (2–7.2)**	**8 (5.2–10)**	**<0.0001 ***
Positive qSOFA, (%)	54.3	20.0	**53.3**	**87.5**	**<0.0001 ***
Lethality, (%)	**28.0**	9.7	**20.0**	**53.1**	**<0.0001 ***
Comorbidities, (%)					
Hypertension	**45.7**	31.3	33.3	**71.9**	**0.001 ***
Diabetes Mellitus	**24.5**	12.5	26.7	34.4	0.115
Cardiopathy	**28.7**	18.8	16.7	**50.0**	**0.007 ***
Nephropathy	12.8	18.8	3.3	15.6	0.164
Hepatopathy	7.4	3.1	13.3	6.3	0.293
Neoplasia	7.4	3.1	13.3	6.3	0.293
Others	21.3	18.8	16.7	28.1	0.542
None	24.5	**37.5**	30.0	6.3	**0.006 ***
Nosocomial infection, (%)	**60.5**	57.7	48.0	73.3	0.156
Focus of infection, (%)					
Unknown	**31.9**	31.3	26.7	37.5	0.673
Abdominal	**14.9**	12.5	16.7	15.6	0.936
Respiratory	**26.6**	15.6	36.7	28.1	0.168
Genitourinary tract	**17.0**	28.1	10.0	12.5	0.158
Skin-soft tissue/bone-joint	6.4	9.4	6.7	3.1	0.687
Catheter	3.2	3.1	3.3	3.1	1.000

Abbreviations: IQR: interquartile range; ICU: intensive care unit; SOFA: Sequential Organ Failure Assessment; qSOFA: quick Sequential Organ Failure Assessment; †: Kruskal–Wallis H test or Fisher’s exact test; *: *p* < 0.05.

**Table 3 biomedicines-12-00933-t003:** Laboratory tests according to diagnosis.

	All Cases (N = 94)	Infection (n = 32)	Sepsis (n = 30)	Septic Shock (n = 32)	*p*-Value †
** median (IQR) **					
Leukocyte (WBCs/mm^3^)	12,780 (8520–17,830)	11,270 (9040–16,010)	14,235 (6647–19,995)	13,060 (7464–21,060)	0.912
Platelet (×10^3^/mm^3^)	224 (130–300)	244 (154–312)	248 (131–409)	182 (108.5–259)	0.184
INR	1.18 (1.03–1.35)	1.08 (1.00–1.21)	**1.18 (1.03–1.40)**	**1.25 (1.13–1.53)**	**0.014 ***
Total bilirubin (mg/dL)	0.90 (0.52–1.79)	0.58 (0.36–1.05)	**0.97 (0.49–5.02)**	**1.11 (0.69–1.82)**	**0.023 ***
C-reactive protein (mg/L)	112.5 (64.4–128.8)	118.1 (50.1–132.0)	111.6 (55.2–127.3)	110.3 (66.1–129.1)	0.972
Creatinine (mg/dL)	1.49 (0.74–2.32)	0.78 (0.63–1.39)	**1.5 (0.78–2.45)**	**1.83 (1.07–3.11)**	**0.012 ***
Lactate (mmol/L)	1.9 (1.2–2.9)	1.6 (1.1–2.5)	**1.7 (1.1–3.6)**	**2.45 (1.8–3.5)**	**0.030 ***

Abbreviations: IQR: interquartile range; INR: international normalized ratio; †: Kruskal–Wallis H test or Fisher’s exact test; *: *p* < 0.05.

**Table 4 biomedicines-12-00933-t004:** Diagnostic performance of cfDNA for assessing the severity of sepsis using ROC curve and AUC analysis in patients with infection, sepsis, and septic shock.

cfDNA	Cut-Off Value **	AUC (95% CI)	Sensitivity (%)	Specificity (%)
L1PA2_90_	18.07	0.817 (0.725–0.909) *	77.0	79.3
L1PA2_222_	18.27	0.741 (0.634–0.849) *	66.6	81.4
cf-mtDNA	38.58	0.703 (0.596–0.810) *	48.2	90.6
cf-nDNA + SOFA	-	0.916 (0.853–0.979) *	88.1	80.0

Abbreviations: ** cut-off values expressed in log2 copy numbers/mL. Abbreviations: ROC: receiver operating characteristic; AUC: area under the curve; 95% CI: confidence interval at the level 95%; * *p* < 0.001.

**Table 5 biomedicines-12-00933-t005:** Performance of cfDNA in predicting transfer to ICU using ROC and AUC analysis in patients with infection, sepsis, and septic shock.

cfDNA	Cut-Off Value **	AUC (95% CI)	Sensitivity z (%)	Specificity (%)
L1PA2_90_	17.17	0.692 (0.559–0.806) *	100.0	34.2
L1PA2_222_	18.11	0.653 (0.520–0.770) *	66.6	62.5
cf-mtDNA	40.17	0.596 (0.463–0.720)	25.0	100.0
cf-nDNA + SOFA	-	0.752 (0.622–0.855) *	95.2	50.0

Abbreviations: ** cut-off values expressed in log2 copy numbers/mL. Abbreviations: ROC: receiver operating characteristic; AUC: area under the curve; 95% CI: confidence interval at the level 95%; * *p* < 0.001.

## Data Availability

The datasets used and/or analyzed during the current study are available from the corresponding author on reasonable request.

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
