# Peer review of "Cell-Free Nuclear and Mitochondrial DNA as Potential Biomarkers for Assessing Sepsis Severity"

_biomedicines, 2024, doi:10.3390/biomedicines12050933_

Round 1

Reviewer 1 Report

Comments and Suggestions for Authors

The details of enrolling patients is lacking. Based on the ratio of the three arms it looks like that there is a risk of section bias. Are the patients enrolled consecutively? A flow chart for patients enrollment and exclusion would be helpful. 

The performance evaluation should be based on biomarker ability in distinguish between normal, inflammation and infection. In the study, however, the authors present biomarker ability in distinguishing infection and sepsis/septic shock. The study has to be redesigned.

Author Response

Dear Ms. Jolie Li,

We want to thank the reviewers for their consideration and comments.

Our gratitude for significant suggests enhancements to our manuscript. We are sure that their input has improved it.  

Please, attached are the responses and files following a detailed correction of the manuscript entitled “Cell-Free Nuclear and Mitochondrial DNA as a Potential Biomarkers for Assessing Sepsis Severity" by Miranda et al., which I am submitting for consideration in Biomedicines.

All changes in the text were highlighted (1) blue: requested by reviewer 1, (2) yellow: requested by reviewer 2 and (3) green: requested by reviewer 3.

Sincerely yours,

Comments and Suggestions for Authors

REVIEWER #1

1) The details of enrolling patients is lacking. Based on the ratio of the three arms it looks like that there is a risk of section bias. Are the patients enrolled consecutively? A flow chart for patients enrollment and exclusion would be helpful.

- We are deeply grateful for the insightful observations shared by the reviewer. A non-probability convenience sampling method was performed, and a total of 131 patients were enrolled. 24 patients were excluded from the study due to missing data in medical records, including the absence of laboratory results, SOFA scores, and outcome information. 15 patients lacked PCR results either due to non-amplification or for not having a plasma sample. Additionally, 2 patients were removed from the study as outliers, identified using the Interquartile Range (IQR) method, resulting in a final sample size of 94 patients. This information has been added to the lines 98 and 111-113.

Flowchart of the patients included in the manuscript.

2) The performance evaluation should be based on biomarker ability in distinguish between normal, inflammation and infection. In the study, however, the authors present biomarker ability in distinguishing infection and sepsis/septic shock. The study has to be redesigned.

We appreciate the reviewer's feedback. Our choice of a control group comprised patients with infections rather than healthy individuals was deliberate to avoid potential overestimation of accuracy results, as highlighted by Whiting et al. (PMID: 23958378) and noted in the STARD 2015 guidelines (PMID: 26511519). This decision was also influenced by the study base, candidate population, and target population. Just as male subjects are not suitable controls in studies of gynecologic conditions and neonates are not appropriate controls in studies of Alzheimer's disease, healthy patients are not at risk for the condition under study and thus not suitable for this study.

To ensure comparability between cases and controls, we selected controls with a similar referral pattern and presumably from the same study base, as well as obtaining a similar quality of information by including patients with clinical and laboratory diagnoses of infection (PMID: 7925728). Wacholder et al. emphasize the importance of cases and controls representing the same underlying base experience, defined as the set of individuals or person-time in which diseased subjects become cases, or the members of the source population for the cases during their eligibility period (PMID: 1595688).

In our manuscript, we explicitly stated that the control group comprises patients with infections to provide clarity. This choice does not diminish the validity of the publication; rather, it enhances the findings by demonstrating the ability to differentiate between patients with infection and those with sepsis/septic shock based on cfDNA. The results are further strengthened when combining cf-nuclear DNA (cf-nDNA) with the Sequential Organ Failure Assessment (SOFA) score.

The incorporation of molecular markers, such as cfDNA, alongside clinical scores like SOFA, can aid physicians in diagnosis by reducing uncertainty and increasing post-test probability (PMID: 24200831 and 22948322).

Reviewer 2 Report

Comments and Suggestions for Authors

Comments on the Quality of English Language

English language editing is suggested

Author Response

Dear Ms. Jolie Li,

We want to thank the reviewers for their consideration and comments.

Our gratitude for significant suggests enhancements to our manuscript. We are sure that their input has improved it.  

Please, attached are the responses and files following a detailed correction of the manuscript entitled “Cell-Free Nuclear and Mitochondrial DNA as a Potential Biomarkers for Assessing Sepsis Severity" by Miranda et al., which I am submitting for consideration in Biomedicines.

All changes in the text were highlighted (1) blue: requested by reviewer 1, (2) yellow: requested by reviewer 2 and (3) green: requested by reviewer 3.

Sincerely yours,

Comments and Suggestions for Authors

REVIEWER #2

The study by Silva de Miranda et al. intends to evaluate the value of cell-free DNA (cf-DNA), both nuclear (cf-nDNA) and mithocondrial (cf-mDNA) for the assessment of the severity and prognosis of sepsis. Furthermore, the Authors analyzed cf-DNA concentrations in patients admitted to ICU. The results obtained would demonstrate that cf-DNA could discriminate patients with infection without sepsis from patients with different sepsis severity.

Major comments:

The study shows several issues that should be considered.

1) In general, the manuscript is not very clear and often confusing, especially in the results. The tables are confusing. It would be advisable to divide homogeneous data either through a grid or into multiple tables.

- We are deeply grateful for the insightful observations shared by the reviewer. To enhance the clarity of table interpretation, we have partitioned Table 1 into three sections and emphasized key themes and values pertinent to the study. This information has been added to the lines 183-193.

Materials and methods

Analytical assays

2) Why do you use the phenol-chloroform method to extract DNA? Breitbach et al. (PLOS One, 2014) suggest to use another method to avoid loss of DNA.

- We appreciate the reviewer’s comment. Since column-based DNA extraction kits may not be as efficient as traditional methods for isolating low quantities of DNA from bodily fluids (PMID: 25826002 and PMID: 11708487), we used a manual, noncolumn based phenol-chloroform method. Furthermore, the cfDNA extraction protocol using phenol-chloroform has a notably longer processing time, but is cost-effective. QIAamp DNA Blood Mini Kit has a high cost, especially in a middle-income country like Brazil, costing around 1000x more per sample. In our laboratory we had already tested QIAamp DNA kits, and our results demonstrated that the phenol-chloroform method had obtained a higher yield (data not show). Other publications demonstrate the similar results (PMID: 22187243 and 19246404).

3) Then, the use of L1PA2 targeting sequences is not well specified. Please, explain better or clearly refer to previous studies. Further, why do you use two different pairs of primers targeting L1PA290 and L1PA2222 to obtain cf-nDNA? This choice is not clear. Please, specify or refer more in detail to already published studies.

- We appreciate the reviewer’s comment. The assay was designed in a way that the forward primer was the same for all nuclear amplicons, whereas the reverse primer varied. Other publications indicated that there might be a difference in the quantity of short versus long fragments, demonstrating differences in the integrity of cfDNA (PMID: 25243646, 28910365, 24876361, and 38015284). The estimated size of cfDNA varies from 40 to 200 bp, with a peak at about 166 bp. Shorter fragments of DNA (<100 bp) have different origins from larger fragments, and cfDNA integrity is determined by the ratio of long to short PCR product amplified from the same locus (such as LINE1 locus). For example, patients with cancer have a significantly elevated level of cfDNA integrity compared to healthy individuals and patients with benign diseases, derived from augmented levels of necrotic death (PMID: 22531347 and 19962739). The same may occur in septic patients since similar cellular mechanisms are observed in these patients (PMID: 19678906).

4) The equations for calculating cf-DNA copy number are unclear. For example, what is the meaning of the number 112,200 in the denominator of the equation for calculating cf-mtDNA? I did not find this number in reference #18.

We appreciate the reviewer's feedback and apologize for the lack of clarity. The equations used to calculate cf-DNA copy number can be found in reference #18 (page 14). The value 112,200 in the denominator of the equation was derived from the formula (2*L*MW) as described by Meddeb et al., 2019.

For cf-mtDNA: mtDNA is the cf-mtDNA copy number per milliliter, ‘c’ is the mtDNA mass concentration (g/µl) determined by a qPCR targeting the mitochondrial gene. NA is Avogadro’s number (6.02 * 1023 molecules per mole), “L” is the mitochondrial amplicon length (85pb) and MW is the molecular weight of one nucleotide (g/mol). Velution is the elution volume of cirDNA extract (µl) and Vplasma is the volume of plasma used for the extraction (ml).

* We used the average molecular mass of one base pair (660 g/mol). 

Thus, the calculation was performed as follows: (2 * 85 bp (mitochondrial amplicon length) * 660 g/mol (MW representing the molecular weight of one nucleotide g/mol) = 112,200.

For cf-nDNA: nDNA is the cf-nDNA copy number per milliliter, “c” nDNA concentration (pg/µl) determined by qPCR targeting the nuclear gene L1PA290 or L1PA2222 sequence and 3.3 pg is the human haploid genome mass. Velution is the volume of cirDNA extract (µl) and Vplasma is the volume of plasma used for the extraction (ml).

We have revised the Materials and Methods section to clarify the equation. This information has been added to the lines 138-153.

Results

As previously specified, results are often confusing and not correctly highlighted.

Major details:

5) Table 1: I suggest separating data by grid and paragraphs in order to homogenize numbers based on their specific characteristics.

- We are deeply grateful for the insightful observations shared by the reviewer. To enhance the clarity of table interpretation, we have partitioned Table 1 into three sections and emphasized key themes and values pertinent to the study as suggested.

6) Furthermore, analysis and correlation with more specific laboratory markers for the diagnosis and prognosis of sepsis is lacking, e.g. PCT, presepsin, MR-proADM. Please, explain why.

- We appreciate the reviewer's feedback. This study was funded by PPSUS (Research program for the Brazilian Unified Health System (SUS)) and was carried out in a Federal Public Hospital with severe financial restrictions. Unfortunately, the Hospital has not yet adopted procalcitonin as a clinical marker due to its high cost for a middle-income country (around 100 dollars per patient), nor did the Project's financing cover other biomarkers (presepsin, MR-proADM), only consumables and materials to carry out molecular analyzes (cfDNA, miRNA and methylation assay).

7) Figure 2: The concentration of total DNA extracted and used for qPCR is not specified. The calculated concentration values should result and the same concentration of DNA should be used for PCR amplification.

We appreciate the reviewer's comment. While quantification of DNA by spectrophotometry is a widely used method in research laboratories, nanodrop spectrophotometry is considered non-specific for cfDNA quantification due to its inability to differentiate between dsDNA and ssDNA, as well as cfDNA fragments and protein-DNA complexes. Therefore, following isolation from plasma, cfDNA was quantified using qPCR, targeting pre-selected DNA sequences. The parameters for qPCR included the plasma volume and elution volume of the sample. In detail, 0,25 ml of plasma each sample was extracted off using the phenol-chloroform method. The final elution volume was 50 µl. Please note that the volume of plasma and elution were used in the equation (see formula). We have revised the Materials and Methods section to clarify. This information has been added to the lines 120-121.

8) The analysis of variance should be performed to demonstrate the homogeneity between the groups, both for demographic characteristics (age and sex) and for the presence of different comorbidities, the mere presence of which could itself increase the quantity of cf-DNA and not sepsis. The correlation should be done among the same pathology (without infection and/or sepsis), cf-DNA increase and sepsis severity.

- We appreciate the reviewer’s comment. We recognize that comorbidities and age are interfering factors in cfDNA quantifications and included them as covariates in our analyses. To demonstrate this more clearly, we added to the methodology that sex, age and comorbidities were added as covariates in our models. This information has been added to the lines 172-173.

9) Figure 3: The scatter plot figure is not clear. Since SOFA score is based on clinical data, how is it expressed numerically in the scatter plots? Please, clarify.

- We appreciate the reviewer’s comment and apologize for the lack of clarity. SOFA evaluates six systems: respiratory, cardiovascular, hepatic, coagulation, renal and neurological, with a score ranging from 0 to 4 for each of these systems, depending on the parameters measured, such as bilirubin, platelets, Glasgow coma scale. In total, their score varies between 0 and 24 points. Thus, SOFA, such as other scores (NEWS 2, APACHE II, SAPS II, qSOFA), is a quantitative score.

10) Further, considering the big dispersion of patients, especially in the cf-mtDNA plotter, it is not easy clearly understand the correlation that the authors intend demonstrate. In fact, it seems that SOFA score alone is able to discriminate. In the ROC curves of Figure 3, the SOFA score and other routinely biomarkers should be analyzed to demonstrate the eventual better accuracy of cf-DNA in respect to the SOFA score and biochemical biomarkers.

- We appreciate the reviewer’s comment. Originally, the Sequential Organ Failure Assessment (SOFA) score was developed to identify organ dysfunction in septic patients. However, it became evident that it is not specific to sepsis, and as a result, it has been adopted for the assessment of critically ill patients regardless of septic status. We recognize there was a large dispersion, especially in septic patients and patients with septic shock, demonstrated by the heat map due to the non-formation of sample clusters, which illustrates the variability of clinical presentations of sepsis.  However, clinical studies demonstrate varying levels of accuracy in sepsis prediction due to its non-specificity, highlighting a diagnostic gap, in which several biomarkers are evaluated to identify their value in the diagnosis and prognosis of this disease (PMID: 30470600, 30477377 and 34991675).

General considerations:

11) The main aim of this study should be to demonstrate the possible use of cf-DNA in clinical diagnostics, but it would be necessary to analyze what advantages could be drawn from this marker compared to markers already used by clinicians, which are certainly more reliable for diagnosis, prognosis and risk stratification of septic patients. As is known, in fact, in sepsis syndrome early diagnosis is essential to avoid the more serious condition of organ failure and septic shock. Therefore, the authors should better explain what could be the advantages of using this type of biomarkers compared to those already in use. The only interest remains exclusively scientific, if well demonstrated, but currently it does not seem helpful in the clinical practice.

- We appreciate the reviewer’s comment. Sepsis remains a persistent challenge in global healthcare, necessitating improved diagnostic and prognostic tools. Despite advancements in treatment, it remains associated with high mortality rates and significant cognitive dysfunction. Extensive research is ongoing to validate biomarkers for sepsis, enabling early intervention. Sepsis exhibits varied inflammatory responses and subsequent immunosuppression, leading to multi-organ dysfunction. Biomarkers offer potential for predicting, identifying, and innovating sepsis treatments.

There exists a pressing need for improved diagnostic tools in sepsis detection and monitoring. Despite various proposed biomarkers, no single indicator has achieved widespread acceptance (PMID: 23108494 and PMID: 20144219). C-reactive protein (CRP) and procalcitonin (PCT) are extensively studied in suspected bacterial sepsis. Both CRP and PCT are today routinely employed in clinical practice but have limited abilities to distinguish bacterial sepsis from other inflammatory conditions (PMID: 18379262, PMID: 27076187, PMID: 18421435 and PMID: 30358811). While PCT is established in sepsis, its the diagnostic accuracy of routine has been questioned due to variable results based on illness severity and infection in the studied patient population (PMID: 24898888, PMID: 26554775). Lactate, another commonly used biomarker, lacks specificity, as elevated levels are not exclusive to sepsis but also occur in conditions like cardiac arrest, trauma, and seizures (PMID: 15190968, PMID: 27387712). Future efforts to distinguish infection from sepsis to septic shock should focus on an expanded clinical-molecular diagnostic panel. Noninvasive analysis of cell-free DNA (cfDNA) holds promise for diverse clinical applications in sepsis management, including diagnosis, immediate treatment response evaluation, longitudinal therapy monitoring, and detection of emerging resistance mechanisms.

Early identification of severe sepsis is crucial for prompt interventions that can potentially reduce mortality rates, emphasizing the clinical utility of prognostic and diagnostic biomarkers in sepsis. Despite blood culture being the gold standard for severe infection diagnosis, its positivity rate in sepsis patients is only around 30%, indicating the need for additional biomarkers.

In the ICU, cfDNA has garnered attention as a potential prognostic and predictive biomarker in sepsis. The hypothesis that cfDNA originates from dead cells and immune cells during sepsis suggests its potential correlation with sepsis severity.

Our findings offer insights into how cfDNA may inform assessments of Sepsis Severity in critically ill patients, aiding clinical decision-making. Larger scale clinical trials are warranted to translate cfDNA profiling assays into practical biomarker applications in the ICU. Moreover, identifying optimal biomarker combinations is essential for enhancing diagnosis, treatment, and patient outcomes. Our study encourages further investigation in this area to advance translational medicine.

Minor comments:

12) The phrase at line 153-154 should be cancelled. - We grateful the reviewer’s note and apologize for the error. The phrase was removed.

13) English editing is suggested.

. - We appreciate the reviewer’s feedback. We submitted the manuscript to a professional English revision service to ensure the article meets the standards expected by reviewers.

Reviewer 3 Report

Comments and Suggestions for Authors

Revie of the manuscript Cell-free Nuclear and Mitochondrial DNA as a potential biomarkers for assessing sepsis severity

The manuscript adresses an important topic- namely a search for biomarkers for the diagnosis of sepsis and for severity stratification. The topic is extremely interesting, as mitochodnrial dysfunction is one of the key features of sepsis and investigations targeted on mitochondrial function evaluation are of great interest.

There are major methodological issues:

1.       First, evaluation of these biomarkers as diagnostic tools using ROC analysis is incorrect in my opinion, as there is no control group. To use a predictive or diagnostic tool, according to STARD criteria, there is a control group without infections or sepsis, to be included. Thus, ROC curve analysis might not be appropiate here and the advice of a statistician is required.

2.       The second methodology is correct- the auhtors describe disease stratification based on these biomarkers- they demonstrate that these novel biomarkers discriminate between patients with infections, sepsis and septic shock.

3.       The sample sizes are modest, but adequate for an association study.

The minor concerns are:

Moderate English language corrections- example page 2 lines 77-79 need rephrasing, line 89 was=were

The introduction has a good structure- are there other similar studies? Which is the novelty? How could we use these biomarkers in practice? Since sepsis and septic shock are mainly clinical diagnosis and the number of patients are very high for those admitted in the hospital? Could such techniques enter clinical practice?

Methods- ethcial committee approval should appear in the begining of the section, not after blood withdrawal

Results and data analysis is comprehensively described

Discussion- page 8- lines 300-305: there are many confounding factors, so initial pathologies are of interest, also, other pathologies should be included in such an analysis as false positives could be encountered- thus, it is mandatory to include control groups, without sepsis or infections, but with other pathologies and/or healthy controls. It is not adequate to analyse only patients with infections.

I stongly advice the researchers to include a control group and follow stard criteria.

Comments on the Quality of English Language

moderate corrections

Author Response

Ms. Jolie Li,

Chef in Editor

MDPI - Biomedicines

Manuscript ID: biomedicines-2841347

Title: Cell-Free Nuclear and Mitochondrial DNA as a Potential Biomarkers for Assessing Sepsis Severity.

Authors: Felipe Silva de Miranda, Livia Maria de Araújo Maia Cláudio, Dayanne Silva Monteiro de Almeida, Juliana Braga Nunes, Valério Garrone Barauna, Wilson Barros Luiz, Paula Frizzera Vassallo, Luciene Cristina Gastalho Campos*

Journal: Biomedicines – Molecular Biomarkers and More Efficient Therapies for Sepsis

Article: Original Research

Submitted on: 9 Jan 2024

Dear Ms. Jolie Li,

We want to thank the reviewers for their consideration and comments.

Our gratitude for significant suggests enhancements to our manuscript. We are sure that their input has improved it.

Please, attached are the responses and files following a detailed correction of the manuscript entitled “Cell-Free Nuclear and Mitochondrial DNA as a Potential Biomarkers for Assessing Sepsis Severity" by Miranda et al., which I am submitting for consideration in Biomedicines.

 All changes in the text were highlighted (1) blue: requested by reviewer 1, (2) yellow: requested by reviewer 2 and (3) green: requested by reviewer 3.

 Sincerely yours,

Comments and Suggestions for Authors

REVIEWER #3

The manuscript adresses an important topic- namely a search for biomarkers for the diagnosis of sepsis and for severity stratification. The topic is extremely interesting, as mitochodnrial dysfunction is one of the key features of sepsis and investigations targeted on mitochondrial function evaluation are of great interest.

There are major methodological issues:

1) First, evaluation of these biomarkers as diagnostic tools using ROC analysis is incorrect in my opinion, as there is no control group. To use a predictive or diagnostic tool, according to STARD criteria, there is a control group without infections or sepsis, to be included. Thus, ROC curve analysis might not be appropiate here and the advice of a statistician is required.

- We acknowledge the reviewer's feedback. Our control group consists of patients with infections. We opted not to include healthy patients to prevent an overestimation of accuracy results, as illustrated by Whiting et al. (PMID: 23958378) and referenced by STARD 2015 itself (PMID: 26511519). We emphasize that the choice of patients with infection as a control group was also based on the study base, the candidate population, and the target population. In the same way that male subjects cannot serve as controls in a study of gynecologic conditions, and neonates cannot serve as controls in a study of Alzheimer's disease, healthy patients are not suitable to be a control group in this study because they are not at risk for the condition under study. To achieve this, we select this controls whose referral pattern is similar to that of the cases and who presumably come from the same study base, and we obtain a similar quality of information when selecting patients with clinical and laboratory diagnosis of infection (PMID: 7925728). Wacholder et al. addresses that cases and controls should be representative of the same base experience. The base is the set of persons or person-time, depending on the context, in which diseased subjects become cases. The base can also be thought of as the members of the underlying cohort or source population for the cases during the time periods when they are eligible to become cases (PMID: 1595688). To make it more understandable, we included in the manuscript methodology that the control group is composed by patients with infection. This information has been added to the lines 99-101.

The second methodology is correct- the authors describe disease stratification based on these biomarkers- they demonstrate that these novel biomarkers discriminate between patients with infections, sepsis and septic shock.

The sample sizes are modest, but adequate for an association study.

The minor concerns are:

2) Moderate English language corrections- example page 2 lines 77-79 need rephrasing, line 89 was=were

. - We appreciate the reviewer’s feedback and apologize for the errors in the English language. We submitted the manuscript to a professional English revision service to ensure the article meets the standards expected by reviewers.

As requested, lines 77-79 have been revised for clarity (now lines 77-81), and in line 89, the verb has been adjusted accordingly (now line 95).

4) The introduction has a good structure- are there other similar studies? Which is the novelty? How could we use these biomarkers in practice? Since sepsis and septic shock are mainly clinical diagnosis and the number of patients are very high for those admitted in the hospital? Could such techniques enter clinical practice?

. - We appreciate the reviewer’s comments. Numerous studies have investigated the predictive value of cell-free DNA (cfDNA) in sepsis-related mortality. Currently, there are at least 20 such studies involving approximately 1,000 patients. Nine of these studies have compared cfDNA levels between survival and non-survival groups. Additionally, 13 studies, comprising 1,000 samples, have examined cfDNA concentrations in sepsis patients versus controls (healthy volunteers), as well as in sepsis versus non-sepsis patients in the Intensive Care Unit (ICU).

The diagnostic potential of cfDNA to differentiate between sepsis and non-sepsis in the ICU has been explored in at least eight studies involving 878 patients. However, there are several other published works that either lack microbiological evidence in their clinical sepsis diagnosis data, provide limited raw data reports for Receiver Operating Characteristic (ROC) curve analysis, or do not compare cfDNA with other sepsis biomarkers.

Our study aimed to investigate circulating cell-free DNA (cfDNA) as potential diagnostic biomarkers for distinguishing infection, sepsis, and septic shock. As novelty we found that cfDNA concentrations increase with the severity of sepsis, particularly when comparing cf-nDNA with ccf-mtDNA. Notably, cf-nDNA effectively differentiated between patients with infection and those with sepsis or septic shock. Combining cf-nDNA with organ dysfunction scores significantly improved accuracy, sensitivity, and specificity, aiding in sepsis diagnosis. This trend was also evident in ICU hospitalization assessments. Elevated levels of cf-nDNA in patients admitted to the ICU after a sepsis diagnosis not only reflect the severity of their clinical condition but also suggest cfDNA's potential as an early indicator for escalated care needs. Overall, cfDNA has the potential to assist healthcare professionals in evaluating illness severity, prognosis, and ICU admission decisions, demonstrating promising clinical applicability. Thus, our data strengthen the role of cfDNA as non-invasive biomarkers for sepsis. Additional investigations involving diverse populations are imperative to authenticate the outcomes attained with cfDNA in sepsis.

cfDNA holds significant promise as a biomarker in clinical practice. However, While the Surviving Sepsis Campaign acknowledges the potential utility of novel biomarkers in sepsis management, the updated guidelines have yet to endorse the use of any specific biomarkers for prognostic or diagnostic purposes in sepsis.

Delays in the management of sepsis, including diagnosis and treatment, have been associated with increased mortality rates, extended hospital stays, and elevated treatment costs. This underscores the necessity for dependable biomarkers that can facilitate early sepsis diagnosis and prognostic prediction. Cell-free DNA (cfDNA) warrants further investigation, particularly through qualitative and quantitative analyses, in comparison with established biomarkers for sepsis diagnosis and prognosis. We propose cfDNA as a potential measure to guide clinical assessments in the identification of sepsis and the prediction of sepsis survival outcomes.

5) Methods- ethcial committee approval should appear in the begining of the section, not after blood withdrawal

- We appreciate the reviewer's feedback. The text has been reorganized, with ethical considerations now placed at the beginning of the section This information has been added to the lines 91-94.

Results and data analysis is comprehensively described

6) Discussion- page 8- lines 300-305: there are many confounding factors, so initial pathologies are of interest, also, other pathologies should be included in such an analysis as false positives could be encountered- thus, it is mandatory to include control groups, without sepsis or infections, but with other pathologies and/or healthy controls. It is not adequate to analyse only patients with infections.

I stongly advice the researchers to include a control group and follow stard criteria.

- We appreciate the reviewer's feedback. As discussed previously (question 1) the control group adopted was patients with infections for the reasons mentioned earlier.

The control group composed with patients diagnosed with infection does not invalidate the manuscript's publication, rather, it strengthens the findings. It illustrates the ability to differentiate between patients with infection and those with sepsis/septic shock based on cfDNA. The results are further improved when cf-nDNA is combined with SOFA. We recognize that cfDNA can originate from other pathologies, which can lead to false-positive results, and that our population of cases and controls have comorbidities and that there are significant differences in distributions between groups (as described in table 1). Of the total of 32 patients with infection, only 12 did not have comorbidities. Of the 62 patients diagnosed with sepsis or septic shock, only 13 had no comorbidities. Therefore, comorbidities were included as covariates in our analyses.

Regarding false positives, our results demonstrate a positive likelihood ratio value of 4.41 (CI95% 2.21-8.92), leading a post-test probability of 89.1%, and resulting in 7 false-positive patients for every 100 pattients tested. False-positive tests are inherent to the diagnosis. We cannot interpret any diagnostic test without estimating the patient's clinical probability. Every medical diagnosis is probabilistic, with different degrees of uncertainty and must be analyzed using Bayes' rule. The adoption of molecular markers, such as cfDNA, in association with clinical scores, such as SOFA, can assist medical doctors in their diagnosis, reducing this degree of uncertainty by increasing post-test probability (PMID: 24200831 and 22948322).

Round 2

Reviewer 1 Report

Comments and Suggestions for Authors

Dear authors,

Thank you for the efforts on the revisions. Please find my comments as belows:

1. "A non-probability convenience sampling method was performed..." While there would be selection-bias in such patients inclusion method, it is necessary to reveal that the patients enrollment was not consecutively and not randomly. Risk of bias may exist.

2. I can accept your argument on the reason why you presented biomarker ability in distinguishing infection and sepsis/septic shock rather than health versus infection. If so, I am not convinced by the motivation and the results of the study. I agree with you that there is no gold standard for sepsis/septic shock. Even Sepsis-3 is just a concensus. While your aim is to identify an useful marker of sepsis/septic shock, you should provide more evidence that the new marker is a better prognostic marker than the traditional approach (i.e. Sepsis-3 based diagnosis). Showing the survival curves stratified by mtDNA versus curves strtified by SOFA would be helpful. I would say showing the diagnostic performance of mtDNA in assessing sepsis severity is not so useful from the clinical aspect. The reason is that current Sepsis-3 approach is easy and fast. When I can rapidly and easily get the Sepsis-3 based sepsis/septic shock diagnosis, what would be reason to spend more money and time to test another marker whose diagnotic performance is suboptimal? Thus, I guess the key is to argue and prove that mtDNA is better than the Sepsis-3 based approach. Evaluating mtDNA on the basis of Sepsis-3 is not likely to win, espeically when the performance of cf-mtDNA is low. It is hard to claim it is a disease marker with the performance level.  

3. Regarding the diagnostic performance level, unfortunately, I would like to say it is more likely a negative finding study. The whole manuscript should be re-written in the opposite tone.

Author Response

Comments and Suggestions for Authors  

REVIEWER #1

  1. "A non-probability convenience sampling method was performed..." While there would be selection-bias in such patients inclusion method, it is necessary to reveal that the patients enrollment was not consecutively and not randomly. Risk of bias may exist.

We appreciate your feedback and apologize for the misunderstanding regarding the patient selection description sent in round 1 comments. We clarify one hundred and thirty-one (131) patients were enrolled consecutive in the study convenient nonprobability sampling. The cohort included patients over 18 years of age suspected with sepsis presenting to the emergency department, admitted to the ICU, and internal medicine in various working shifts. Sepsis diagnosis was confirmed after clinical examinations, meeting the required criteria Sepsis-3 consensus definitions and performance of laboratory tests. Patients with a prior history of cancer, individuals in a palliative care state, pregnant women, those who declined to provide consent for study participation, subjects for whom a sepsis diagnosis could not be confirmed were excluded from this study. The study spanned from 2017 to 2018, during which 131 consecutive patients met the inclusion criteria, with 37 individuals excluded due to incomplete data. A comprehensive outline of patient selection criteria and exclusion parameters can be found in lines 99 to 103 of the manuscript.

Considering the bias mentioned, it is worth highlighting that biases are inherent in scientific studies and can lead to underestimation or overestimation of results. It is important to acknowledge and address these biases to ensure the accuracy and reliability of study findings. The reviewer's concern regarding selection bias in patient inclusion methods is valid, and we have taken steps to address this issue. Additional information has been included in the manuscript to clarify that patient enrollment was consecutive convenient nonprobability sampling. Hung et al. Biomedicines 2020 (PMID: 33198109) in their scientific publication highlight the importance of considering biases when interpreting results from clinical studies. These biases can arise from various sources, including participant selection, methodology employed, and data analysis methods. Is important to emphasize that all studies, including this one, are subject to biases.

Furthermore, multiple studies are essential in the quest for potential biomarkers for the diagnosis and monitoring of sepsis due to the complex nature of this clinical disease. Sepsis is characterized by dysregulated immune responses that can persist long after initial recovery, leading to immunological suppression, and persistent inflammation, which are often implicated in patient mortality. While advancements in treatment protocols recommended by the Surviving Sepsis Campaign have reduced mortality rates over the past decade, they remain unacceptably high. Identifying at-risk patients before organ dysfunction sets in is crucial, necessitating rapid diagnosis and immediate treatment initiation.

Traditional culture methods for sepsis diagnosis are time-consuming, lacking a gold standard. Early diagnosis and effective management are paramount in reducing sepsis mortality. However, the vast physiological variability in infection response and the nonspecific nature of sepsis symptoms present challenges to early diagnosis. To address this challenge, recent research has focused on searching for biomarkers that hold promise in predicting sepsis diagnosis, prognosis, and treatment response monitoring. Hence, numerous studies have explored potential biomarkers to improve sepsis diagnosis, reflecting the urgent need for reliable diagnostic tools in sepsis management.

So, it is worth noting the positive aspects of this study, such as the inclusion of patients from diverse hospital sectors, the use of controls composed of patients with infections, and the demonstration of improved results by combining cfDNA with existing diagnostic scores. These aspects contribute to the robustness of the results and suggest that cfDNA may be a promising tool for early identification of sepsis and its complications. Our results are encouraging and indicate significant potential for future investigations and developments in the detection and monitoring of sepsis using cfDNA as a biomarker.

  1. I can accept your argument on the reason why you presented biomarker ability in distinguishing infection and sepsis/septic shock rather than health versus infection. If so, I am not convinced by the motivation and the results of the study. I agree with you that there is no gold standard for sepsis/septic shock. Even Sepsis-3 is just a concensus. While your aim is to identify an useful marker of sepsis/septic shock, you should provide more evidence that the new marker is a better prognostic marker than the traditional approach (i.e. Sepsis-3 based diagnosis). Showing the survival curves stratified by mtDNA versus curves strtified by SOFA would be helpful. I would say showing the diagnostic performance of mtDNA in assessing sepsis severity is not so useful from the clinical aspect. The reason is that current Sepsis-3 approach is easy and fast. When I can rapidly and easily get the Sepsis-3 based sepsis/septic shock diagnosis, what would be reason to spend more money and time to test another marker whose diagnotic performance is suboptimal? Thus, I guess the key is to argue and prove that mtDNA is better than the Sepsis-3 based approach. Evaluating mtDNA on the basis of Sepsis-3 is not likely to win, espeically when the performance of cf-mtDNA is low. It is hard to claim it is a disease marker with the performance level.

We express sincere gratitude for the valuable feedback provided by the reviewer. However, as previously described, our deliberate selection of a control group comprising patients with infections rather than healthy individuals was intended to avoid potential overestimation of accuracy results, as highlighted by Whiting et al. (PMID: 23958378) and emphasized in the STARD 2015 guidelines (PMID: 26511519). This decision was also guided by the study's rationale, the characteristics of the candidate population, and the intended target population.

It is essential to highlight that healthy individuals were not included in our study, as they do not present a risk for the investigated condition, making them unsuitable for this research. Furthermore, it was not our intention to perform a comparison of survival curves, as this would divert us from our main objective of identifying sepsis and assessing its severity.

Similar to the preceding discourse elucidated in question 1, sepsis biomarkers have attracted significant attention in scientific research, driven by advances in scientific understanding and the quest to overcome knowledge barriers, as highlighted in the study by Awirut Charoensappakit et al. in Scientific Reports 2023 (PMID: 37949942).

Our research group has been conducting studies in the area of bridging the gap between fundamental biomedical research and healthcare innovation. We highlight, that our goal is the identification of sepsis and its severity, not the study of 28-day mortality. We also emphasize that our combined data do not advocate replacing the current sepsis-3 diagnostic algorithm or clinical criteria; instead, they provide valuable information about strategies to improve sepsis identification and monitor its clinical progression. Specifically, our findings suggest that cfDNA can complement existing approaches and markers in clinical practice.

Furthermore, our data indicate that the use of these markers could facilitate early decision-making. Given the diverse financial and technical scenarios of hospitals, the integration of these markers could have substantial potential. Moreover, leveraging technological advances, such as artificial intelligence tools, can improve identification models, leading to tangible benefits for patients. Given the complexities of the sepsis response, panels of biomarkers or models combining biomarkers and clinical data are necessary, along with specific data analysis methods falling under the scope of machine learning, as discussed in the work by Matthieu Komorowski et al. in Lancet 2022 (PMID: 36470834).

As previously discussed, it is crucial to reinforce that biomarker research in sepsis is vital, with significant challenges in this area, as reviewed by Marshall and Leligdowicz in EBioMedicine 2022 (PMID: 36470831).

Finally, it is imperative to underscore the significance of translational research, personalized medicine, patient stratification, and early identification of intervention needs, particularly in the context of sepsis, characterized by diverse clinical presentations and lacking a gold standard in diagnosis. Citing the PMID articles: 27978985 and 37949942 further reinforces these critical aspects.

  1. Regarding the diagnostic performance level, unfortunately, I would like to say it is more likely a negative finding study. The whole manuscript should be re-written in the opposite tone.

We grateful the reviewer. However, our results demonstrated an AUC of 0.916 (CI95 %: 0.853-0.979), with a sensitivity of 88.1 % and specificity of 80.0 % in identifying septic patients. With a positive likelihood ratio value of 4.41 (CI95 %: 2.21-8.92), the post-test probability was 89.1 %, resulting in 7 false positives and 8 false negatives for every 100 patients tested. This shows that the adoption of molecular markers, such as cfDNA, in association with clinical scores, such as SOFA, can assist medical doctors in their diagnosis. Sepsis still has a diagnostic gap that is explored by translational research in which several biomarkers are evaluated to identify their value in the diagnosis and prognosis of sepsis (PMID: 36470831). A potentially useful biomarker of sepsis is not simply one that correlates with the presence of the syndrome, but rather one that in a heterogeneous population of patients, identifies a subpopulation that is more likely to benefit from a particular treatment approach, as showed by our study in which cfDNA identified patients who are likely to be admitted to the ICU with a sensitivity of 95.2 %. cfDNA can support medical doctors by predicting or detecting complications of sepsis. Although all results require an analytical validation to ensures the consistency of cfDNA with a multicenter study of a larger population to confirm the clinical usefulness of cfDNA for detection of sepsis and their complications. However, as demonstrated in a cpouple reviews by Barichello (PMID: 34991675), Liu (PMID: 28028489) and Pierrakos (PMID: 32503670), different biomarkers have been evaluated in sepsis, such as acute phase proteins, cytokines and chemokines, endothelial cells and BBB markers, non-coding RNAs, membrane receptors, cell proteins, cfDNA, etc. Each one with different degrees of accuracy, varying the AUC between 0.224 and 0.987. Therefore, we believe that our results are not negative, and demonstrate greater accuracy than most of 200 biomarkers listed by Barichello, Pierrakos and Liu, even among biomarkers already used in clinical practice such as procalcitonin and C-reactive protein. Extensive research in the area is being performed to validate biomarkers, facilitate sepsis diagnosis, and allow an early intervention that, although primarily supportive, can reduce the risk of death. In the future, biomarkers with better diagnostic value and combined diagnosis with multiple biomarkers or clinical scores are expected to solve the challenge of the diagnosis of sepsis.

Reviewer 2 Report

Comments and Suggestions for Authors

N/A

Author Response

No requests were made by the reviewer during round 2 of manuscript review.

Reviewer 3 Report

Comments and Suggestions for Authors

The authors have adressed main concerns regarding the manuscript and I consider it is suitable to be considered for publication.

Author Response

(The authors gave the same response as above.)
